# COVID-19 Management in a UK NHS Foundation Trust with a High Consequence Infectious Diseases Centre: A Retrospective Analysis

**DOI:** 10.3390/medsci9010006

**Published:** 2021-02-04

**Authors:** Kenneth F. Baker, Aidan T. Hanrath, Ina Schim van der Loeff, Su Ann Tee, Richard Capstick, Gabriella Marchitelli, Ang Li, Andrew Barr, Alsafi Eid, Sajeel Ahmed, Dalvir Bajwa, Omer Mohammed, Neil Alderson, Clare Lendrem, Dennis W. Lendrem, Lucia Pareja-Cebrian, Andrew Welch, Joanne Field, Brendan A. I. Payne, Yusri Taha, David A. Price, Christopher Gibbins, Matthias L. Schmid, Ewan Hunter, Christopher J. A. Duncan

**Affiliations:** 1Translational and Clinical Research Institute, Newcastle University, Newcastle upon Tyne NE2 4HH, UK; kenneth.baker@ncl.ac.uk (K.F.B.); Aidan.Hanrath@newcastle.ac.uk (A.T.H.); Ina.Schim-van-der-Loeff@newcastle.ac.uk (I.S.v.d.L.); brendan.payne@nhs.net (B.A.I.P.); 2National Institute of Health Research (NIHR) Newcastle Biomedical Research Centre, Newcastle University and The Newcastle upon Tyne Hospitals NHS Foundation Trust, Newcastle upon Tyne NE2 4HH, UK; dennis.lendrem@newcastle.ac.uk; 3The Newcastle upon Tyne Hospitals NHS Foundation Trust, Newcastle upon Tyne, Newcastle upon Tyne NE1 4LP, UK; suann.tee@nhs.net (S.A.T.); richard.capstick@nhs.net (R.C.); gabriella.marchitelli@nhs.net (G.M.); ang.li1@nhs.net (A.L.); andrew.barr6@nhs.net (A.B.); alsafi.eid@nhs.net (A.E.); sajeel.ahmed@nhs.net (S.A.); dalvir.bajwa1@nhs.net (D.B.); omer.mohammed1@nhs.net (O.M.); n.alderson@nhs.net (N.A.); lucia.pareja-cebrian@nhs.net (L.P.-C.); andrew.welch1@nhs.net (A.W.); joanne.field@nhs.net (J.F.); yusri.taha@nhs.net (Y.T.); david.price15@nhs.net (D.A.P.); christopher.gibbins1@nhs.net (C.G.); matthias.schmid1@nhs.net (M.L.S.); ewan.hunter1@nhs.net (E.H.); 4NIHR In Vitro Diagnostics Cooperative, Newcastle University, Newcastle upon Tyne NE2 4HH, UK; clare.lendrem@newcastle.ac.uk

**Keywords:** treatment escalation, ventilation, intubation, SARS-CoV-2, mortality

## Abstract

Recent large national and international cohorts describe the baseline characteristics and outcome of hospitalised patients with COVID-19, however there is limited granularity to these reports. We aimed to provide a detailed description of a UK COVID-19 cohort, focusing on management and outcome. We performed a retrospective single-centre analysis of clinical management and 28-day outcomes of consecutive adult inpatients with SARS-CoV-2 PCR-confirmed COVID-19 from 31 January to 16 April 2020 inclusive. In total, 316 cases were identified. Most patients were elderly (median age 75) with multiple comorbidities. One quarter were admitted from residential or nursing care. Mortality was 84 out of 316 (26.6%). Most deaths occurred in patients in whom a ceiling of inpatient treatment had been determined and for whom end of life care and specialist palliative care input was provided where appropriate. No deaths occurred in patients aged under 56 years. Decisions to initiate respiratory support were individualised after consideration of patient wishes, premorbid frailty and comorbidities. In total, 59 (18%) patients were admitted to intensive care, of which 31 (10% overall cohort) required intubation. Multiple logistic regression identified associations between death and age, frailty, and disease severity, with age as the most significant factor (odds ratio 1.07 [95% CI 1.03–1.10] per year increase, *p* < 0.001). These findings provide important clinical context to outcome data. Mortality was associated with increasing age. Most deaths were anticipated and occurred in patients with advance decisions on ceilings of treatment.

## 1. Introduction

The first two patients with COVID-19 in the United Kingdom (UK) received inpatient care at The Newcastle upon Tyne Hospitals NHS Foundation Trust, one of five airborne High Consequence Infectious Diseases (HCID) centres, on 31 January 2020 [1]. Since then, 3,689,746 patients have tested positive for SARS-CoV-2 in the UK and 100,162 people have died of COVID-19 (as of 26 January 2021) [2]. Studies from China [3], Spain [4], UK [5], and USA [6] have reported mortality rates of 21–33% of hospitalised patients, but differences in population demographics, health behaviours, and systems of healthcare between these countries may influence both outcome and how outcomes are recorded.

During the ongoing pandemic, clinicians are faced with challenging clinical decisions around appropriate ceilings of treatment, however information regarding this key aspect of COVID-19 management is currently lacking. Such decisions may impact death rates and influence our understanding of factors associated with adverse outcome. We sought to provide a comprehensive description of clinical management pathways, treatment escalation decisions and outcome in a UK COVID-19 inpatient cohort managed in a large NHS Foundation Trust. The goal of this analysis was to provide clinical context to inform crude mortality data across healthcare systems.

## 2. Materials and Methods

### 2.1. Setting

The Newcastle upon Tyne Hospitals NHS Foundation Trust (NUTH) is a large tertiary academic medical centre serving the population of Newcastle upon Tyne (estimated 302,820) [7] and the wider North East of England. In accordance with NHS England guidelines, combined nose and throat swabs or sputum samples were obtained for SARS-CoV-2 reverse-transcriptase polymerase chain reaction (PCR) for all patients hospitalised with symptoms of COVID-19. Initially PCR testing was performed using the Public Health England RdRp assay until 7th April, followed by the Altona Diagnostics (from 1st April) and Roche cobas 6800 (from 7th April) assay platforms.

Of note, all patients were admitted prior to the first randomised clinical trials demonstrating the efficacy of remdesivir and immunomodulatory therapies in severe COVID-19. Indeed, owing to initial concern regarding the potential for corticosteroids to dampen a protective immune response against the SARS-CoV-2 virus, our internal Trust guidelines recommended against the use of corticosteroids in routine COVID-19 clinical care outside of clinical trials. As such, no clinical therapy beyond best supportive care was routinely available to patients during this initial wave of COVID-19 hospitalisation.

### 2.2. Data Collection

We searched our electronic health records to identify all consecutive patients admitted to NUTH between 31 January to 16 April 2020 inclusive with a positive SARS-CoV-2 PCR result. Of 362 patients identified, 46 were excluded: 14 had no relevant admission at the time of testing, 21 were already inpatients at the time of infection, six were below 18 years of age, and five were asymptomatic and screened for an unrelated reason. In total, six patients were inpatient transfers with COVID-19 from other hospitals for whom first recorded observations were not available. Electronic health records were retrospectively reviewed by a team of medical doctors with the aid of a standardised version-controlled data collection template (Excel, Microsoft Corporation) with internal data validation restrictions. Baseline clinical, demographic, and laboratory factors were collected. Comorbidities were defined as clinician reported. Clinical Frailty Scale [8] was calculated retrospectively. Radiological findings were classified according to the British Society of Thoracic Imaging criteria [9], as documented by the reporting radiologist. Severe COVID-19 at admission was defined with reference to the World Health Organisation (WHO) definition of severe COVID-19 disease [10] (respiratory rate > 30 breaths/min and/or oxygen saturations <90% on room air) and/or a new requirement for supplemental oxygen. Clinical status (death, respiratory pressure support, oxygen therapy, or discharged alive) as per the WHO ordinal scale for COVID-19 [11] was recorded for each calendar day for a total of 28 days after hospital admission. All inpatient and outpatient deaths occurring up to this censor date were also recorded, the latter via daily system updates from primary care. Collected data were merged and reviewed for errors (0.8%) and missing data (A.T.H., I.S.v.d.L., and K.F.B.) prior to analysis.

### 2.3. Data Analysis

Analyses were performed in R (version 3.6.0, R Core Team, R Foundation for Statistical Computing, Vienna, Austria) and SAS JMP Pro (version 13.2.1, SAS Institute, Cary, NC, USA). The significance of departure of observed male sex proportion from an expected value of 0.5 was assessed using the one sample z-test. Tests of differences in proportions (χ^2^ test, or Fisher’s exact test where contingency table cell counts ≤ 5) and continuous data (Wilcoxon rank sum test) were performed between contrast groups where stated. Odds ratios for death were calculated between death and survival groups by logistic regression. The sensitivity of odds-ratios to other key variables was explored by constructing a multiple logistic regression examining the impact upon estimates of odds-ratios using iterative step-in and step-out procedures. Note that unlike stepwise regression the candidate for retention is tested in the presence of all subsets of the remaining candidates and interest centres on those that remain statistically significant in all subsets. In addition, a regularized lasso regression [12] using the Bayesian information criterion (BIC) model criterion permitted identification of stable parameters through plots of parameter estimates as a function of model complexity. A two-tailed α < 0.05 was considered statistically significant.

### 2.4. Ethics

The study was registered as a clinical service evaluation with the Newcastle upon Tyne Hospitals NHS Foundation Trust and was exempt from ethical approval, with analysis of anonymised healthcare data approved by the Caldicott Guardian.

## 3. Results

### 3.1. Clinical Features at Presentation

A total of 316 patients were identified with a median (IQR: interquartile range) (range) age of 75 (60–83) (23–101) years (Table 1; Online Appendix A). 281/303 (93%) patients were white British, white Irish, or other white ethnicity. Over half the cohort (55%) was male. Males were disproportionately represented at all ages under 70 years (75 out of 124 (61%), *p* = 0.019). Interestingly, broadly similar proportions of men and women aged 70 years and over were admitted (98 out of 192 (51%) male, *p* = 0.510). Overall, 27 of 316 (9%) patients were healthcare workers.

Median (IQR) (range) symptom duration prior to admission was 5 days (2–9) (0–42). The most common presenting symptoms were cough (224 (71%)), fever (211 (67%)), and breathlessness (197 (62%)), with 286 (91%) patients presenting with at least one of these symptoms. 60 (19%) patients were admitted from a residential or nursing home. In total, 253 out of 316 (80%) patients had at least one major comorbidity, the most common of which were hypertension (133 (42%)), chronic kidney disease (77 (24%)), ischaemic heart disease (65 (21%)), and dementia (55 (17%)).

Overall, 136 out of 308 (44%) patients presented with severe COVID-19, defined according to World Health Organisation (WHO) criteria [10] (respiratory rate > 30 breaths/min and oxygen saturations <90% on room air) and/or a new requirement for supplemental oxygen. In a minority of cases (8 out of 316) data were insufficient to determine WHO severity at presentation. A similar proportion of men and women had severe disease at presentation (71 out of 167 (43%) men, 65 out of 141 (46%) women, *p* = 0.606). Most patients had elevated acute phase reactants on admission, with median (IQR) (range) C-reactive protein (CRP) of 72 (30–131) (<5–523) mg/dL. Lymphopaenia (<1 × 10^9^/L) was observed in 186 out of 311 (60%) patients, and 246 out of 305 (81%) were eosinopaenic (<0.04 × 10^9^/L). In those who underwent testing, elevations were noted in lactate dehydrogenase (94 out of 116, 81%), D-dimer (19 out of 33, 58%), troponin-I (28 out of 52, 53%) and creatine kinase (48 out of 141, 34%).

### 3.2. Patient Outcomes

Daily clinical status (i.e., death, intubation, non-invasive pressure support, or supplemental oxygen) was recorded for 28 days after index admission in all patients. In total, 28 (9%) patients were hospitalised more than once (i.e., readmitted following previous discharge) in the 28 days following date of index admission, and the outcome of the readmission was counted in the analysis. Deaths occurring either in hospital or in the community within 28 days of index admission were recorded for all patients. Overall, 31 (10%) patients remained hospitalised 28 days after their index admission—for these patients, final outcome beyond 28 days (i.e., in-hospital death or survival to discharge) was recorded.

Overall, 84 (27%) patients died, 232 (73%) survived, and one patient remained hospitalised at the time of analysis (138 days after admission). Discounting this remaining inpatient, the median (IQR) (range) duration of hospital admission was 8 (4–14) (1–105) days. Non-respiratory complications possibly associated with COVID-19 occurred in 51 (16%) patients, including cardiac dysrhythmias (24 (8%)), heart failure (11 (3%)), enterocolitis (10 (3%)), stroke (11 (3%)), pulmonary embolus (6 (2%)), and limb ischaemia (3 (1%)). Of note there were no recorded deep vein thromboses.

An epidemic curve showing the daily incidence of admissions, ventilatory support, and mortality for the entire cohort is shown in Figure 1.

### 3.3. Management and Escalation of Care

Decisions on treatment strategies and escalation of care were made on an individual patient basis by clinical teams in consultation with specialist intensive care physicians, with consideration of pre-morbid clinical state and the views of the patient and family and carers. This approach was in line with national UK COVID-19 guidance supporting the use of clinical frailty score as part of holistic individualised assessments to guide decisions on escalation to critical care [13]. Retrospectively, we discerned four distinct patient ‘pathways’: treatment in the intensive care unit (ICU) (‘ICU group’: 59 (19%) patients); non-invasive pressure support (NIPS) at the ward level as recently described [14] (‘ward NIPS group’: 32 (10%) patients); ward-based care (‘standard ward care group’: 176 (56%) patients); and end-of-life care without prior NIPS or ICU admission (‘palliative group’: 49 (16%) patients) (Figure 2). High flow nasal oxygen (HFNO) was infrequently used (19 out of 316 (6%) patients): 15 in the ICU group, 3 in the ward NIPS group, and 1 in the standard care ward group.

The majority of patients received standard ward care (Table 2). Patients in this group were of relatively advanced age (median (IQR) 73 (58–82) years), with a median (IQR) clinical frailty score (CFS) of 4 (2–6), indicating a vulnerable population with substantial premorbid functional limitation. In total, three quarters had an underlying major comorbidity, defined as presence of at least one of: respiratory comorbidity (asthma, COPD, interstitial lung disease, or obstructive sleep apnoea), heart failure, diabetes, active cancer, or immunosuppression. Furthermore, 26 out of 176 (15%) patients lived in a nursing or residential home. The characteristics of this group reflect the advanced age and frailty of our cohort; nevertheless, mortality was low in this group (4 out of 176 (2%)).

As expected, patients who were admitted to ICU were more likely to have severe COVID-19 at presentation compared to those receiving standard ward care (41 out of 54 (76%) vs. 48 out of 173 (28%), *p* < 0.001). 51 out of 59 (86%) of those admitted to ICU required respiratory support, with 31 (53%) requiring intubation and mechanical ventilation and 20 (34%) receiving NIPS only. 15 out of 59 (25%) received HFNO, all of whom received HFNO as an adjunct prior to and/or after NIPS and intubation. Within ICU, vasopressors and renal replacement therapy were required for 27 out of 59 (46%) and 9 out of 59 (15%) patients, respectively. Compared to the standard ward care group, ICU patients were younger (median (IQR) age 60 (53–69), *p* <0.001) and less frail (median (IQR) CFS 2 (2-3), *p* < 0.001), though with similar rates of major comorbidity (39 out of 59 (66%), *p* = 0.246). Compared to the standard ward care group, a significantly greater proportion of patients admitted to ICU were men (41 out of 59 [69%] ICU group vs. 91 out of 176 [52%] standard ward care group, *p* = 0.026). However, it is likely that this is in part reflective of the greater proportion of men admitted with COVID-19 in this younger age group. Both mortality (ICU management 14 out of 58 [24%] vs. standard ward care 4 out of 176 [2%], *p* < 0.001) and length of inpatient stay (median (IQR) 16 (9–26) vs. 7 (3–11) days, *p* < 0.001) were significantly greater in those requiring ICU management versus those managed on the ward (one patient in the ICU group remained hospitalised at the point of analysis). Of those managed in ICU, mortality was higher in patients who received mechanical ventilation versus those who did not (12 out of 31 (39%) vs. 2 out of 28 (7%), *p* = 0.006). There was a statistically non-significant trend towards greater mortality in men admitted to ICU (11 out of 40 (28%) men vs 3 out of 18 (17%) women, *p* = 0.513), though the small number of patients in this group limits further analysis.

A small proportion of patients received NIPS on the ward level without escalation to ICU. These patients had similar rates of severe COVID-19 at presentation (20 out of 32 (63%) vs. 41 out of 54 (76%), *p* = 0.280). This strategy was offered to patients who required respiratory support but in whom ICU admission was not considered in their best interests by the treating clinicians [14]. Patients who received ward-based NIPS were older (median (IQR) age 80 (75–84) years, *p* < 0.001), frailer (median (IQR) CFS 5 (3–6), *p* < 0.001), and a greater proportion had major comorbidities (29/32 (91%), *p* = 0.011) than those managed in ICU. Mortality was significantly greater in the ward-based NIPS group compared to the ICU group (17 out of 32 (53%) vs. 14 out of 58 (24%), *p* = 0.011).

Finally, a group of patients was identified that received active management of end of life based on a policy developed by a multidisciplinary group of infectious diseases, respiratory, care of the elderly and hospital specialist palliative care clinicians. These patients were older and more frail than the other groups (median [IQR] age 85 [79–92], *p* < 0.003 vs all other groups; median (IQR) CFS 6 (5–7), *p* < 0.001 vs all other groups), and two thirds (31 out of 49 (63%)) lived in a nursing or residential home. The decision to initiate end-of-life care was documented by a senior clinician with discussion with the patient (where possible) and their relatives, and 44 out of 49 (90%) were reviewed by a member of the hospital specialist palliative care team. A total of 46 out of 49 patients died in hospital, two were discharged with an end-of-life care plan and died in the community, and one patient recovered and survived to discharge. In line with the end of life infection control policy enacted in NUTH, close relatives could visit prior to death and were provided with appropriate personal protective equipment. On-call chaplaincy and other spiritual support was available throughout the study period.

During admission all patients received thromboprophylaxis with high-dose low molecular weight heparin (LMWH), unless clinically contraindicated. Upon discharge thromboprophylaxis, with low-dose rivaroxaban (10 mg once daily), was continued for 28 days.

### 3.4. Association of Baseline Factors with Mortality

The association between baseline factors at the point of admission and mortality was investigated by univariate logistic regression. Increasing age was strongly associated with mortality (Figure 3), with a median age of 82 years in those who died versus 69 years in those who survived (OR 1.08 per year (95% CI 1.06–1.11), *p* < 0.001). No deaths occurred in those aged under 56 years. Multiple additional factors at presentation were associated with increased mortality, including absence of fever as a presenting symptom; pre-existing heart failure, hypertension, or dementia; increased clinical frailty score; hypoxia; raised respiratory rate; reduced renal function; elevated CRP; severe COVID-19 [10]; and higher CURB65 score [15] (Online Appendix A).

We selected variables to include as prospective candidates in a multiple logistic regression satisfying all of the following criteria: unambiguous to obtain from retrospective data, demonstrating a clinically meaningful difference, unlikely (based on clinical knowledge) to be a proxy measure for another variable in the dataset, identified as a risk factor in other cohorts, and with sufficient available baseline data. Based on these criteria, we selected seven key variables: age, male sex, severe COVID-19, CRP, estimated glomerular filtration rate (eGFR), clinical frailty score, and the presence of one or more major comorbidity. CURB65 was not included as several factors contributing to it were individually included or part of severe disease criteria. CRP was selected in place of neutrophils. We performed multiple logistic regression analyses for patients with complete data for all variables (*n* = 294) exploring the sensitivity of the solutions to the inclusion or exclusion of the other candidates. The full model included age, male sex, severe COVID-19, CRP, eGFR, clinical frailty score, and the presence of one or more major comorbidity. There were robust associations between mortality and just three of the variables: age (OR 1.07 (95% CI 1.03–1.10) per year increase, *p* < 0.001), clinical frailty score (OR 1.31 (1.08–1.59) per one point increase, *p* = 0.006) and severe COVID-19 at presentation (OR 2.43 (1.26–4.66), *p* = 0.008). In addition, there were statistically significant effects for male sex and CRP though these were marginal (*p* = 0.031 and *p* = 0.042, respectively) and sensitive to the inclusion or exclusion of other candidates.

## 4. Discussion

Outcomes in our cohort broadly reflect national experience. Overall mortality was 27%, compared to 33% in the UK-based International Severe Acute Respiratory and emerging Infections Consortium (ISARIC) cohort [5], and similar numbers in US (21%) [6] and China (28%) [3] datasets. There are many possible factors that might influence differences in crude mortality rates between and within countries, including admission policy, demographics, disease severity in those admitted, testing criteria, and inpatient management. The ISARIC data reveal an older population of hospitalised patients in the UK compared to other countries, with a greater burden of comorbidity and imply the possibility of advanced decision making about ceiling of treatment; however, granular detail was lacking.

Our data shed light on the clinical decisions occurring both in general wards and in critical care settings. Whilst admission to critical care in our cohort (19%) was comparable to national (17%) [5] and international (7–26%) [3,4,6] experience, there was also evidence of advanced decision making for individual patients. This had an impact on mortality rates. Most patients in our cohort died in the group in whom a documented ceiling of treatment plan was discussed in advance with the patient and their family and appropriate end of life care was instituted. These patients were frailer, had a higher number of comorbidities, and were more likely to reside in a residential or nursing home, implying that their premorbid risk of COVID-19 mortality was high [16]. Data from the USA indicate extremely high mortality rates in this group despite invasive ventilation [17,18]. Similarly, a fifth of patients who died received ward-based NIPS for single organ (respiratory) failure where there was an advance decision not to escalate to mechanical ventilation in the event of a failure to respond. Thus death was anticipated in most patients dying in hospital in our cohort. Palliative care teams provided specialist input into their management. Integral to our patient-centred approach was to implement a policy of permitting a single visit from relatives, using personal protective equipment, to patients at the end of life.

Mortality in patients receiving mechanical ventilation (39%) compared favourably to the UK experience from ISARIC (53%) [5], and to early international reports of extremely high rates of death (97%) in ventilated patients [3]. Rates of mechanical ventilation were 497 out of 6628 (7%) in the ISARIC cohort and 31 out of 316 (10%) in our cohort, which appears lower than in US (20%) [6] and Chinese cohorts (17%) [3]. A possible interpretation is that mechanical ventilation has been applied more selectively in the UK to a hospitalised population with high rates of frailty and comorbidity—an approach that is consistent with national guidance for critical care management of patients with COVID-19 [13]. In our cohort, NIPS was widely used, either as the ceiling of treatment in patients with respiratory failure and for whom escalation to mechanical ventilation was not considered appropriate [14], or as a bridge to mechanical ventilation in critical care and on medical wards. Emerging clinical trial data support a mortality benefit of NIPS beyond standard oxygen therapy in COVID-19 management, though the optimal patient group(s) and timing of initiation remain to be identified [19].

We observed a disproportionately greater number of men admitted with COVID-19 at all ages under 70 years. Furthermore, significantly more men were admitted to ICU than women, despite similar disease severity at presentation. This is in part due to the greater number of men in the younger ICU group, though does suggest a greater burden of COVID-19 illness amongst men than women in our cohort. This is in keeping with similar observations in national [5] and international [4,6] studies. We also observe trends towards greater mortality in men in the ICU group and in the cohort as a whole, although these fall short of robust statistical significance, presumably due to the relatively small numbers of patients.

Baseline characteristics of our cohort reflect the broader UK experience and contrast with international experience. Our cohort had a median age of 75 and included very few patients under 40 (6%). Most patients had one or more comorbidities and approximately one fifth of our cohort lived in a residential or nursing home. The proportion of patients with severe disease on admission was also high (44%), reflecting the national admission policy across the NHS. Nevertheless, there were also differences. Nosocomial infections made up a low proportion of total cases (21 out of 362 (6%)) and relatively few patients were healthcare workers (27 out of 316 (9%)) or people of Black, Asian, and minority ethnic (BAME) background (7%). In total, eight patients were admitted when management of COVID-19 occurred exclusively in HCID units in order to prevent community spread; these patients were younger, with mild disease, and all survived.

Multiple logistic regression of baseline clinical factors that were associated with death highlighted age, frailty, and disease severity as statistically significant factors. Of these, age had the most significant association. This is a consistent finding worldwide [3,20,21], confirming that regional demographic variations are likely to have a major impact on mortality. We observed no deaths in patients under 56 years of age, although overall numbers in this group were relatively small (56 out of 316 (18%)). Furthermore, the association between death and frailty observed in our cohort has also been consistently reported in other studies across a range of international healthcare systems [22,23,24]. We also examined the influence of ethnicity, finding no evidence of an association. Since there are fewer people from BAME backgrounds in the North East of England compared to other regions, nothing can be inferred from these findings. We were unable to analyse the influence of obesity, which was identified as a factor in the ISARIC cohort [5], due to missing data on height and weight in this acutely unwell cohort (we considered estimates of body mass index (BMI) in clinical notes to be unreliable). We observed a relatively low rate of thromboembolic disease compared to other reports. It is not possible to determine whether this was associated with the clinical policy of universal thromboprophylaxis.

This study has several limitations. Data were retrospectively collected at a single NHS Trust, and may therefore not reflect COVID-19 transmission patterns in other parts of the UK nor necessarily reflect inpatient management across the wider UK NHS. Whilst our cohort size is similar to published analyses [3,25], the number of patients is relatively low. In addition, the modelling was based on a subset of patients for which adequate data was available and excluded those with nosocomial infection. Strengths of this analysis are the extended length of follow up, which is longer than most published cohorts, definite clinical endpoint data in all but one patient, and robust clinical informatics mechanisms to capture deaths in the community occurring after discharge.

This report provides a detailed description of the inpatient management of COVID-19 at the individual patient level, complementing and enriching existing literature and helping to provide context to crude mortality data. These results will be informative to clinicians, policy makers, and healthcare providers.

## Figures and Tables

**Figure 1 medsci-09-00006-f001:**
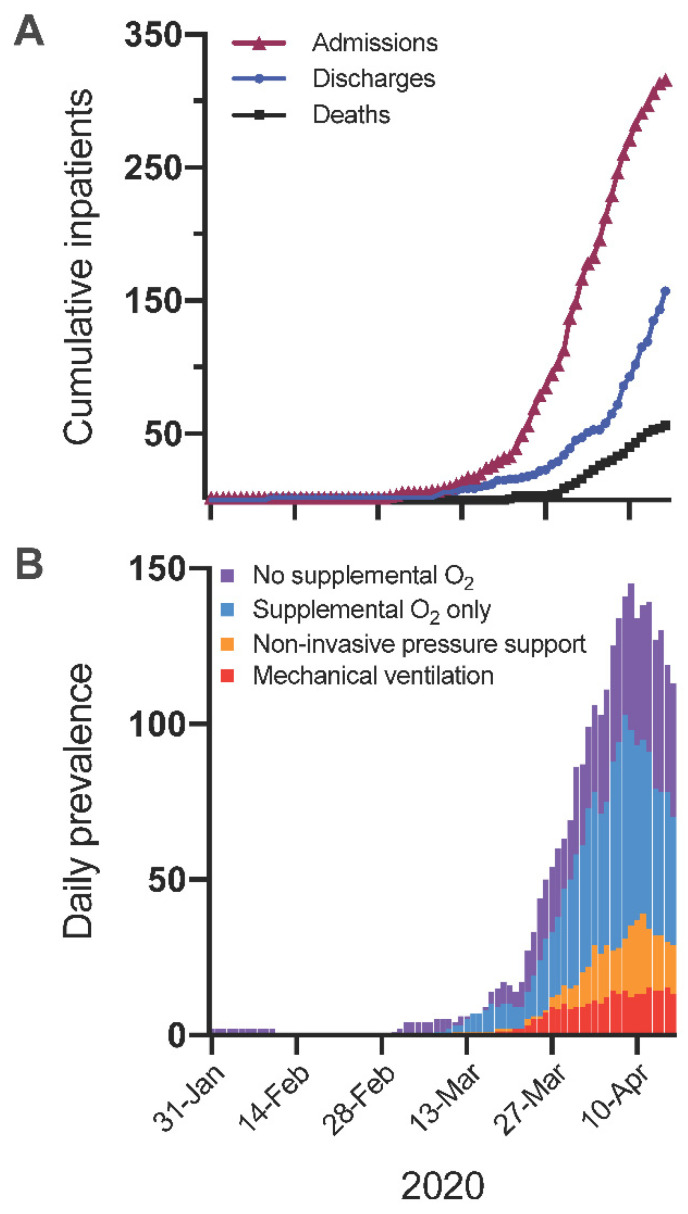
(**A**): Cumulative daily incidence of admissions, discharges and deaths up to censor point of 16 April 2020. (**B)**: Daily prevalence of inpatients with COVID-19 by oxygen and ventilation requirements up to censor point of 16 April 2020.

**Figure 2 medsci-09-00006-f002:**
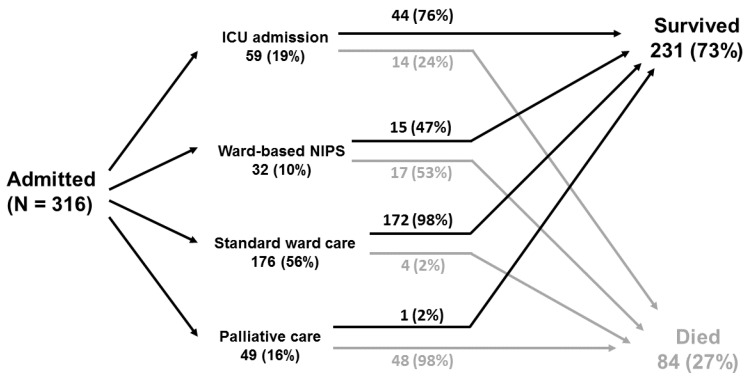
Schematic summarising treatment groups and patient outcomes. Note one patient in ICU group remained hospitalised at time of analysis. ICU: intensive care unit; NIPS: non-invasive pressure support.

**Figure 3 medsci-09-00006-f003:**
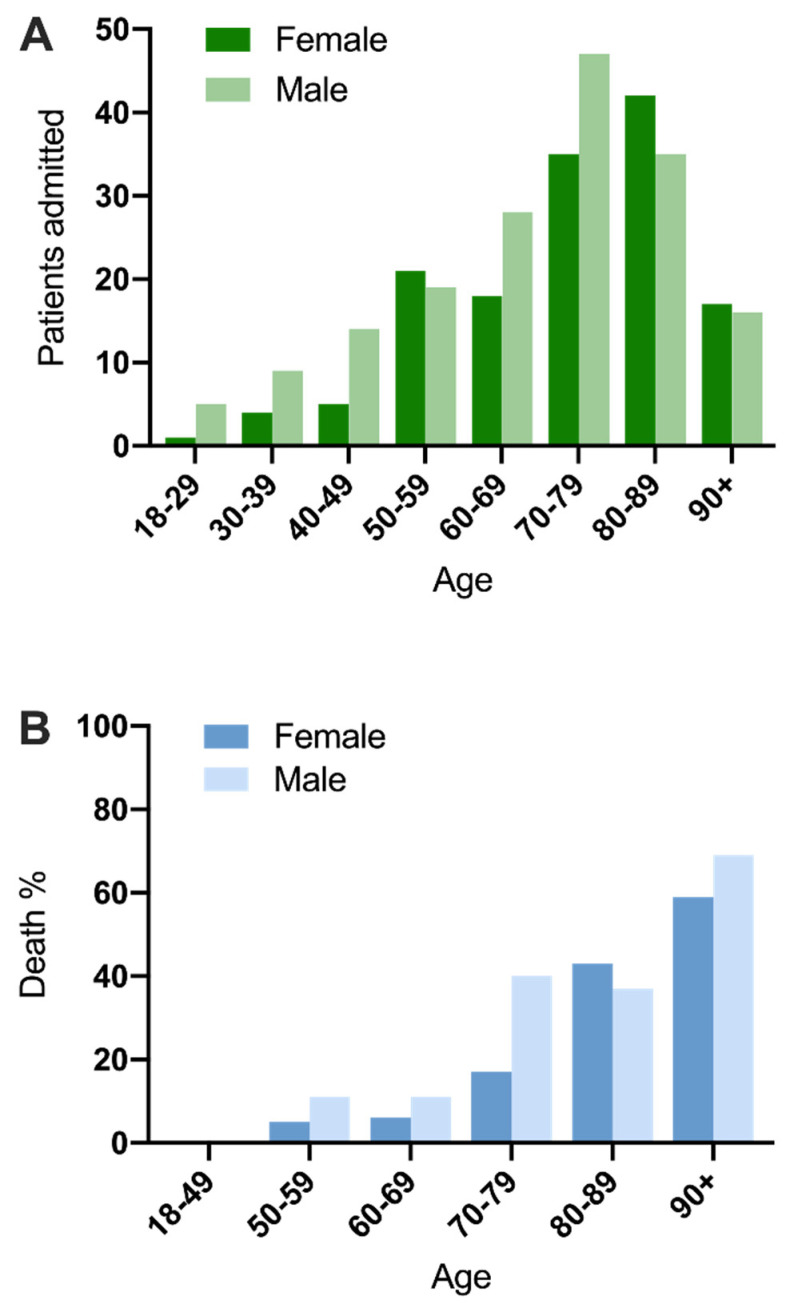
(**A**): Admissions by age and sex of cohort of 316 patients. (**B**): Percentage of deaths among 315 patients with a confirmed clinical outcome, stratified by age and sex.

**Table 1 medsci-09-00006-t001:** Baseline cohort characteristics including univariate logistic regression analyses and associated *p* values. Odds ratios are stated per 1 unit increase in the independent variable unless otherwise stated.

	N	Cohort (316)	Survived (231)	Died (84)	Univariate OR_death_ (95% CI)	*p*
Age (years)	316	75 (60–83)	69 (56–80)	82 (76–89)	1.08 (1.06–1.11)	<0.001
Male sex	316	173 (54.7)	124 (53.6)	48 (57.1)	1.15 (0.70–1.91)	0.585
Fever	316	211 (66.8)	165 (71.4)	46 (54.8)	0.48 (0.29–0.81)	0.006
Cough	316	224 (70.9)	169 (73.2)	55 (65.5)	0.70 (0.41–1.20)	0.184
Sputum	316	76 (24.1)	63 (27.3)	13 (15.5)	0.49 (0.24–0.92)	0.033
Breathlessness	316	197 (62.3)	136 (58.9)	61 (72.6)	1.85 (1.08–3.25)	0.027
Any comorbidity ^a^	316	250 (79.1)	171 (74.0)	78 (92.9)	4.56 (2.03–12.2)	0.001
Respiratory comorbidity ^b^	316	101 (32.0)	78 (33.8)	23 (27.4)	0.74 (0.42–1.27)	0.284
Immunosuppression ^c^	316	27 (8.5)	17 (7.4)	9 (10.7)	1.51 (0.62–3.46)	0.341
Dementia	316	55 (17.4)	29 (12.6)	26 (31.0)	3.12 (1.70–5.73)	<0.001
Healthcare worker	316	27 (8.5)	27 (11.7)	0 (0)	na	na
Admitted from nursing or residential home	316	60 (19.0)	27 (11.7)	33 (39.3)	4.89 (2.71–8.92)	<0.001
Clinical frailty score	311	4 (2–6)	3 (2–5)	6 (4–7)	1.61 (1.40–1.87)	<0.001
Severe COVID-19	308	174/308 (56.5%)	85/225 (37.8)	51/83 (61.4)	2.62 (1.57–4.44)	<0.001
Haemoglobin (115–165 g/L)	310	134 (115–145)	135 (117–145)	130 (115–145)	0.95 (0.84–1.07) ^f^	0.390
Lymphocytes (1–4 × 10^9^/L)	311	0.88 (0.64–1.32)	0.92 (0.65–1.33)	0.82 (0.62–1.32)	0.99 (0.87–1.07)	0.882
Neutrophils (2–7 × 10^9^/L)	311	5.30 (3.70–7.48)	5.01 (3.54–7.09)	7.24 (4.37–9.30)	1.17 (1.08–1.26)	<0.001
eGFR (>90 mL/min.1.73 m^2^)	307	68 (44–89)	76 (53–>90)	52 (32–74)	0.88 (0.83–0.92) ^g^	<0.001
CRP (<5 mg/L)	306	72 (30–131)	65 (23–119)	90 (50–176)	1.04 (1.01–1.06) ^f^	0.007
Hypoxia ^d^	308	192/308 (62.3)	129/225 (57.3)	63/83 (75.9)	2.75 (1.64–4.64)	<0.001
Definite COVID-19 on baseline CXR	303	121/303 (39.9)	88/219 (40.2)	32/83 (38.6)	0.93 (0.55–1.56)	0.796
CURB65 score ^e^	299	2 (1–2)	1 (0–2)	2 (1–3)	2.20 (1.67–2.95)	<0.001

Data are median (IQR: interquartile range) for continuous variables, and n (%) (or n/N (%)) for categorical variables. Local laboratory normal ranges for blood tests are shown in parentheses. ^a^ Presence of at least one of respiratory comorbidity, heart failure, diabetes, active cancer or immunosuppression. ^b^ Defined as at least one of: asthma, chronic obstructive pulmonary disease (COPD), interstitial lung disease, obstructive sleep apnoea, home nebuliser/oxygen/non-invasive pressure support. ^c^ Defined as at least one of: immunodeficiency syndrome, maintenance steroids (prednisolone ≥ 5 mg/day, hydrocortisone ≥ 15 mg/day, any dose dexamethasone); conventional synthetic immunosuppressive drugs (excluding hydroxychloroquine and sulfasalazine); biologics; Janus kinase (JAK) inhibitors; cytotoxic chemotherapy within past 6 months. ^d^ Defined as oxygen saturations ≤ 94% on room air, or any use of supplemental oxygen. ^e^ CURB65 with 1 point each for confusion, urea > 7, respiratory rate > 30, systolic blood pressure < 90 mmHg or 60 mmHg diastolic, and age ≥ 65. ^f^ OR stated per 10 unit increase in haemoglobin/CRP. ^g^ OR stated per 5 unit increase in eGFR. CRP: C-reactive protein, COPD: chronic obstructive pulmonary disease, CXR: chest x-ray, eGFR: estimated glomerular filtration rate, IQR: interquartile range, N: total number of measurements for each variable for the entire cohort, OR: odds ratio.

**Table 2 medsci-09-00006-t002:** Patient characteristics stratified by treatment group. Data are median (IQR) {range} for continuous variables, and *n*. (%) (or *n*/N (%)) for categorical variables.

Characteristic	ICU Admission (*n* = 59)	Ward-Based NIPS (*n* = 32)	Standard Ward Care (*n* = 176)	Palliative Care (*n* = 49)
Age (years)	60 (53–69) {27–80}	80 (75–84) {44–91}	73 (58–82) {23–101}	85 (79–92) {63–97}
Male sex	41 (69)	15 (47)	91 (52)	26 (53)
Severe COVID-19 at presentation	41/54 (76)	20 (63)	48/173 (28)	27 (55)
Clinical Frailty Score	2 (2–3) {1–5}	5 (3–6) {1–7}	4 (2–6) {1–8}	6 (5–7) {4–9}
Admitted from nursing/residential home	0 (0)	3 (9)	26 (15)	31 (63)
Any comorbidity	39 (66)	29 (91)	132 (75)	45 (92)
Admission duration in days	16 (9–26) {1–84}	9 (6–15) {2–105}	7 (3–11) {1–60}	7 (4–11) {1–32}
Mortality	14/58 (24)	17 (53)	4 (2)	48 (98)

## Data Availability

In accordance with UK data protection regulations, the authors are not permitted to share individualised patient data.

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
