# Peer review of "COVID-19 Management in a UK NHS Foundation Trust with a High Consequence Infectious Diseases Centre: A Retrospective Analysis"

_medsci, 2021, doi:10.3390/medsci9010006_

Round 1
Reviewer 1 Report
This is a retrospective analysis COVID-19 management in a UK NHS Foundation Trust with a High Consequence Infectious Diseases centre.
In general it's well written but I'm missing some novelty in their data. What the authors have published is published elsewhere and I think that particularly the data for frailty might be highlighted
I am missing some of the definitions used in methods such as severe COVID. What does it mean?
I am missing a more detailed explanation of the multivariate analysis. It only says that you did a multivariate analysis but not which method ( enter, stepwise,...) and also which variables were selected in the model and how you did select those
I am missing information if patients in ICU were ventilated or not and the mortality
I'm missing the info about the use of corticosteroids in the population overall and in ICU
No information about the use of HFNO
Author Response
Comment 1: In general it's well written but I'm missing some novelty in their data. What the authors have published is published elsewhere and I think that particularly the data for frailty might be highlighted
Response 1: Thank you. We agree that studies examining the association of frailty with mortality deserve highlighting, and have added this to the discussion, together with additional references (Discussion, page 12, lines 340-341).
Comment 2: I am missing some of the definitions used in methods such as severe COVID. What does it mean?
Response 2: We defined severe COVID-19 with reference to the World Health Organisation definition for severe COVID-19 disease, namely a respiratory rate > 30 breaths/min and/or oxygen saturations <90% on room air. In some cases, patients were admitted to hospital by ambulance already receiving oxygen therapy, and oxygen saturations on room air at admission prior to initiation of oxygen therapy were not available in the medical records. To account for this, we also included any use of supplemental oxygen on admission within our definition of severe disease. This definition of severe disease, which we previously described only in the results section, has now also been included in the methods section, and has been made more explicit (Methods and Materials 2.2, pages 2-3, lines 90-93; Results 3.1, page 3, lines 135-136).
Comment 3: I am missing a more detailed explanation of the multivariate analysis. It only says that you did a multivariate analysis but not which method ( enter, stepwise,...) and also which variables were selected in the model and how you did select those.
Response 3: Thank you for your comment. We have added clarification to make it explicit that we selected seven variables for inclusion in the multiple logistic regression (age, male sex, severe COVID-19, CRP, eGFR, clinical frailty score, and the presence of one or more major comorbidity) (Results 3.4, page 11, lines 273-275). We selected these variables based on the following criteria: unambiguous to obtain from retrospective data, demonstrating a clinically meaningful difference, unlikely (based on clinical knowledge) to be a proxy measure for another variable in the dataset, identified as a risk factor in other cohorts, and with sufficient available baseline data (Results 3.4, pages 10-11, lines 264-268). We performed multiple logistic regression analyses to explore the sensitivity of models to the stepwise addition or exclusion of all other candidate variables. Age, clinical frailty score, and severe COVID-19 disease were the only variables which demonstrated associations robust to the stepwise inclusion and exclusion of other candidate variables (Results section 3.4, page 11, lines 275-277).
Comment 4: I am missing information if patients in ICU were ventilated or not and the mortality
Response 4: 31/59 (53%) of patients admitted to ICU required intubation (Abstract, page 1, line 37; Results section 3.3, page 9, lines 210-211). Of those managed in ICU, mortality was higher in patients who received mechanical ventilation versus those who did not (12/31 [39%] vs. 2/28 [7%], p = 0.006) (Results section 3.3, page 9, lines 223-224).
Comment 5: I'm missing the info about the use of corticosteroids in the population overall and in ICU
Response 5: Thank you for highlighting this. In this manuscript we provide data relating to the management and outcomes of patients admitted during the first wave of COVID-19 infection in the UK during early 2020. The patients in this cohort were admitted 2-3 months prior to the first report of the positive effects of corticosteroids in COVID-19 by the RECOVERY trial. At the time of this analysis steroids were not used in the inpatient management of COVID-19 outside of a clinical trial setting. A small number of patients (approx. 20-30) were enrolled into the RECOVERY or REMAP-CAP trials, which included dexamethasone as one of several arms. We have added an explanation of this to the manuscript in order to provide this important contextual information on available contemporaneous treatments (Materials and Methods 2.1, page 2, lines 70-76)
Comment 6: No information about the use of HFNO
Response 6: Thank you for suggesting this. High-flow nasal oxygen was used infrequently within our setting (19/316 [6%] patients): 15 in the ICU group, 3 in the ward NIPS group, and 1 in the standard care ward group. Of the 15 patients who received HFNO in ICU, all received HFNO as an adjunct prior to and/or after NIPS and/or intubation. This data has been added to section 3.2 of the manuscript (page 8, lines 187-188, and page 9, lines 211-212).
Reviewer 2 Report
This description of a covid-19 cohort in England during the early phase of the pandemic including demographics and clinical findings is well-written and of special interest as the virus is new to humans. Detailed descriptions of cohorts are valuable, especially demographics as it is still unknown how different groups are hit differently. This study starts out with showing sex disaggregated data but this perspective is then absent which is my major concern with this paper. How many of the persons admitted to ICU are women? how are the clincial findings among women and men? We miss thus out of potentially interesting biological characteristics and the analysis should be simple. Another question is if some of the antihypertensiv drugs are actually protective as recent data show so the types of drugs would be interesting to see.
if these concerns could be adjusted for it would be fine but I do suggest this publication is published due to the current interest.
Author Response
This description of a covid-19 cohort in England during the early phase of the pandemic including demographics and clinical findings is well-written and of special interest as the virus is new to humans. Detailed descriptions of cohorts are valuable, especially demographics as it is still unknown how different groups are hit differently.
Comment 1: This study starts out with showing sex disaggregated data but this perspective is then absent which is my major concern with this paper. How many of the persons admitted to ICU are women? how are the clincial findings among women and men? We miss thus out of potentially interesting biological characteristics and the analysis should be simple.
Response 1: Thank you for your positive comments, and for your comments above. We agree that further detail of sex disaggregated data and discussion of this would strengthen the manuscript. Sex-specific data and discussion in the revised manuscript is now presented as follows:
- We have added the proportions of men and women admitted with severe disease (71/167 [43%] men, 65/141 [46%] women, p = 0.606) (Results 3.1 page 3, lines 137-138)
- Male sex proportions for each treatment group is already listed in Table 2, and has been further highlighted in additional text in Results 3.3 (page 9, lines 216-219)
- Sex-specific mortality in ICU has been added - 11/40 [28%] men vs 3/18 [17%] women, p = 0.513 (Results 3.3, page 9, lines 224-227)
- Sex-specific mortality in the overall cohort is already presented in Figure 3B, in Table 1, and in Results 3.4 (page 11, lines 277-279).
- We have added a short paragraph on the effect of male sex on outcomes in the Discussion section (page 12, lines 317-324)
Comment 2: Another question is if some of the antihypertensiv drugs are actually protective as recent data show so the types of drugs would be interesting to see.
Response 2: We agree that the recent observations regarding the protective or deleterious effects of specific antihypertensive drugs in COVID-19 deserve further study. However, we are limited by the retrospective nature of this study, and the small sample size for specific drug treatment subgroups. In Supplementary Table S1, we already present the proportion of patients admitted on concurrent ACE inhibitor/ ARBs therapy, and find no significant association with mortality in a univariate logistic regression (ORdeath 0.87 (95% CI 0.47 - 1.54), p = 0.632). Owing to the limitations above, we unfortunately do not have sufficient data to further analyse the effects of specific antihypertensive medications on COVID-19 outcomes.
Round 2
Reviewer 1 Report
It can be published